# HONOR: Hybrid Optimization for NOn-convex Regularized problems

**Pinghua Gong**
Univeristy of Michigan, Ann Arbor, MI 48109
`gongp@umich.edu`

**Jieping Ye**
Univeristy of Michigan, Ann Arbor, MI 48109
`jpye@umich.edu`

## Abstract

Recent years have witnessed the superiority of non-convex sparse learning formulations over their convex counterparts in both theory and practice. However, due to the non-convexity and non-smoothness of the regularizer, how to efficiently solve the non-convex optimization problem for large-scale data is still quite challenging. In this paper, we propose an efficient Hybrid Optimization algorithm for NOn-convex Regularized problems (HONOR). Specifically, we develop a hybrid scheme which effectively integrates a Quasi-Newton (QN) step and a Gradient Descent (GD) step. Our contributions are as follows: (1) HONOR incorporates the second-order information to greatly speed up the convergence, while it avoids solving a regularized quadratic programming and only involves matrix-vector multiplications without explicitly forming the inverse Hessian matrix. (2) We establish a rigorous convergence analysis for HONOR, which shows that convergence is guaranteed even for non-convex problems, while it is typically challenging to analyze the convergence for non-convex problems. (3) We conduct empirical studies on large-scale data sets and results demonstrate that HONOR converges significantly faster than state-of-the-art algorithms.

## 1 Introduction

Sparse learning with convex regularization has been successfully applied to a wide range of applications including marker genes identification [19], face recognition [22], image restoration [2], text corpora understanding [9] and radar imaging [20]. However, it has been shown recently that many convex sparse learning formulations are inferior to their non-convex counterparts in both theory and practice [27, 12, 23, 25, 16, 26, 24, 11]. Popular non-convex sparsity-inducing penalties include Smoothly Clipped Absolute Deviation (SCAD) [10], Log-Sum Penalty (LSP) [6] and Minimax Concave Penalty (MCP) [23]. Although non-convex sparse learning reveals its advantage over the convex one, it remains a challenge to develop an efficient algorithm to solve the non-convex optimization problem especially for large-scale data.

DC programming [21] is a popular approach to solve non-convex problems whose objective functions can be expressed as the difference of two convex functions. However, a potentially non-trivial convex subproblem is required to solve at each iteration, which is not practical for large-scale problems. SparseNet [16] can solve a least squares problem with a non-convex penalty. At each step, SparseNet solves a univariate subproblem with a non-convex penalty which admits a closed-form solution. However, to establish the convergence analysis, the parameter of the non-convex penalty is required to be restricted to some interval such that the univariate subproblem (with a non-convex penalty) is convex. Moreover, it is quite challenging to extend SparseNet to non-convex problems with a non-least-squares loss, as the univariate subproblem generally does not admit a closed-form solution. The GIST algorithm [14] can solve a class of non-convex regularized problems by iteratively solving a possibly non-convex proximal operator problem, which in turn admits a closed-form solution. However, GIST does not well exploit the second-order information. The DC-PN algorithm

[18] can incorporate the second-order information to solve non-convex regularized problems but it requires to solve a non-trivial regularized quadratic subproblem at each iteration.

In this paper, we propose an efficient Hybrid Optimization algorithm for NOn-convex Regularized problems (HONOR), which incorporates the second-order information to speed up the convergence. HONOR adopts a hybrid optimization scheme which chooses either a Quasi-Newton (QN) step or a Gradient Descent (GD) step per iteration mainly depending on whether an iterate has very small components. If an iterate does not have any small component, the QN-step is adopted, which uses L-BFGS to exploit the second-order information. The key advantage of the QN-step is that it does not need to solve a regularized quadratic programming and only involves matrix-vector multiplications without explicitly forming the inverse Hessian matrix. If an iterate has small components, we switch to a GD-step. Our detailed theoretical analysis sheds light on the effect of such a hybrid scheme on the convergence of the algorithm. Specifically, we provide a rigorous convergence analysis for HONOR, which shows that every limit point of the sequence generated by HONOR is a Clarke critical point. It is worth noting that the convergence analysis for a non-convex problem is typically much more challenging than the convex one, because many important properties for a convex problem may not hold for non-convex problems. Empirical studies are also conducted on large-scale data sets which include up to millions of samples and features; results demonstrate that HONOR converges significantly faster than state-of-the-art algorithms.

## 2 Non-convex Sparse Learning

We focus on the following non-convex regularized optimization problem:

$$\min_{\mathbf{x} \in \mathbb{R}^n} \left\{ f(\mathbf{x}) = l(\mathbf{x}) + r(\mathbf{x}) \right\}, \tag{1}$$

where we make the following assumptions throughout the paper:

(A1) $l(\mathbf{x})$ is coercive, continuously differentiable and $\nabla l(\mathbf{x})$ is Lipschitz continuous with constant $L$. Moreover, $l(\mathbf{x}) > -\infty$ for all $\mathbf{x} \in \mathbb{R}^n$.

(A2) $r(\mathbf{x}) = \sum_{i=1}^{n} \rho(|x_i|)$, where $\rho(t)$ is non-decreasing, continuously differentiable and concave with respect to $t$ in $[0, \infty)$; $\rho(0) = 0$ and $\rho'(0) \neq 0$ with $\rho'(t) = \partial \rho(t)/\partial t$ denoting the derivative of $\rho(t)$ at the point $t$.

**Remark 1** *Assumption (A1) allows $l(\mathbf{x})$ to be non-convex. Assumption (A2) implies that $\rho(|x_i|)$ is generally non-convex with respect to $x_i$ and the only convex case is $\rho(|x_i|) = \lambda|x_i|$ with $\lambda > 0$. Moreover, $\rho(|x_i|)$ is continuously differentiable with respect to $x_i$ in $(-\infty, 0) \cup (0, \infty)$ and non-differentiable at $x_i = 0$. In particular, $\partial \rho(|x_i|)/\partial x_i = \sigma(x_i)\rho'(|x_i|)$ for any $x_i \neq 0$, where $\sigma(x_i) = 1$, if $x_i > 0$; $\sigma(x_i) = -1$, if $x_i < 0$ and $\sigma(x_i) = 0$, otherwise. In addition, $\rho'(0) > 0$ must hold (Otherwise $\rho'(0) < 0$ implies $\rho(t) \leq \rho(0) + \rho'(0)t < 0$ for any $t > 0$, contradicting the fact that $\rho(t)$ is non-decreasing). It is also easy to show that, under the assumptions above, both $l(\mathbf{x})$ and $r(\mathbf{x})$ are locally Lipschitz continuous. Thus, the Clarke subdifferential [7] is well-defined.*

The commonly used least squares loss and the logistic regression loss satisfy the assumption (A1); we can add a small term $\delta\|\mathbf{x}\|^2$ to make them coercive. The following popular non-convex regularizers satisfy the assumption (A2), where $\lambda > 0$ and $\theta > 0$ except that $\theta > 2$ for SCAD.

- LSP: $\rho(|x_i|) = \lambda \log(1 + |x_i|/\theta)$.

- SCAD: $\rho(|x_i|) = \begin{cases} \lambda|x_i|, & \text{if } |x_i| \leq \lambda, \\ \frac{-x_i^2 + 2\theta\lambda|x_i| - \lambda^2}{2(\theta-1)}, & \text{if } \lambda < |x_i| \leq \theta\lambda, \\ (\theta+1)\lambda^2/2, & \text{if } |x_i| > \theta\lambda. \end{cases}$

- MCP: $\rho(|x_i|) = \begin{cases} \lambda|x_i| - x_i^2/(2\theta), & \text{if } |x_i| \leq \theta\lambda, \\ \theta\lambda^2/2, & \text{if } |x_i| > \theta\lambda. \end{cases}$

Due to the non-convexity and non-differentiability of problem (1), the traditional subdifferential concept for the convex optimization is not applicable here. Thus, we use the Clarke subdifferential [7] to characterize the optimality of problem (1). We say $\bar{\mathbf{x}}$ is a Clarke critical point of problem (1), if $\mathbf{0} \in \partial^o f(\bar{\mathbf{x}})$, where $\partial^o f(\bar{\mathbf{x}})$ is the Clarke subdifferential of $f(\mathbf{x})$ at $\mathbf{x} = \bar{\mathbf{x}}$. To be self-contained,

we briefly review the Clarke subdifferential: for a locally Lipschitz continuous function $f(\mathbf{x})$, the Clarke generalized directional derivative of $f(\mathbf{x})$ at $\mathbf{x} = \bar{\mathbf{x}}$ along the direction $\mathbf{d}$ is defined as

$$f^o(\bar{\mathbf{x}}; \mathbf{d}) = \limsup_{\mathbf{x} \to \bar{\mathbf{x}}, \alpha \downarrow 0} \frac{f(\mathbf{x} + \alpha \mathbf{d}) - f(\mathbf{x})}{\alpha}.$$

Then, the Clarke subdifferential of $f(\mathbf{x})$ at $\mathbf{x} = \bar{\mathbf{x}}$ is defined as

$$\partial^o f(\bar{\mathbf{x}}) = \{\boldsymbol{\delta} \in \mathbb{R}^n : f^o(\bar{\mathbf{x}}; \mathbf{d}) \geq \mathbf{d}^T \boldsymbol{\delta}, \forall \mathbf{d} \in \mathbb{R}^n\}.$$

Interested readers may refer to Proposition 4 in the Supplement A for more properties about the Clarke Subdifferential. We want to emphasize that some basic properties of the subdifferential of a convex function may not hold for the Clarke Subdifferential of a non-convex function.

## 3 Proposed Optimization Algorithm: HONOR

Since each decomposable component function of the regularizer is only non-differentiable at the origin, the objective function is differentiable, if the segment between any two consecutive iterates do not cross any axis. This motivates us to design an algorithm which can keep the current iterate in the same orthant of the previous iterate. Before we present the detailed HONOR algorithm, we introduce two functions as follows:

Define a function $\pi : \mathbb{R}^n \mapsto \mathbb{R}^n$ with the $i$-th entry being:

$$\pi_i(x_i; y_i) = \begin{cases} x_i, & \text{if } \sigma(x_i) = \sigma(y_i), \\ 0, & \text{otherwise}, \end{cases}$$

where $\mathbf{y} \in \mathbb{R}^n$ ($y_i$ is the $i$-th entry of $\mathbf{y}$) is the parameter of the function $\pi$; $\sigma(\cdot)$ is the sign function defined as follows: $\sigma(x_i) = 1$, if $x_i > 0$; $\sigma(x_i) = -1$, if $x_i < 0$ and $\sigma(x_i) = 0$, otherwise.

Define the pseudo-gradient $\diamond f(\mathbf{x})$ whose $i$-th entry is given by:

$$\diamond_i f(\mathbf{x}) = \begin{cases} \nabla_i l(\mathbf{x}) + \rho'(|x_i|), & \text{if } x_i > 0, \\ \nabla_i l(\mathbf{x}) - \rho'(|x_i|), & \text{if } x_i < 0, \\ \nabla_i l(\mathbf{x}) + \rho'(0), & \text{if } x_i = 0, \ \nabla_i l(\mathbf{x}) + \rho'(0) < 0, \\ \nabla_i l(\mathbf{x}) - \rho'(0), & \text{if } x_i = 0, \ \nabla_i l(\mathbf{x}) - \rho'(0) > 0, \\ 0, & \text{otherwise}, \end{cases}$$

where $\rho'(t)$ is the derivative of $\rho(t)$ at the point $t$.

**Remark 2** *If $r(\mathbf{x})$ is convex, $\diamond f(\mathbf{x})$ is the minimum-norm sub-gradient of $f(\mathbf{x})$ at $\mathbf{x}$. Thus, $-\diamond f(\mathbf{x})$ is a descent direction. However, $\diamond f(\mathbf{x})$ is not even a sub-gradient of $f(\mathbf{x})$ if $r(\mathbf{x})$ is non-convex. This indicates that some obvious concepts and properties for a convex problem may not hold in the non-convex case. Thus, it is significantly more challenging to develop and analyze algorithms for a non-convex problem.*

Interestingly, we can still show that $\mathbf{v}^k = -\diamond f(\mathbf{x}^k)$ is a descent direction at the point $\mathbf{x}^k$ (refer to Supplement D and replace $\mathbf{p}^k = \pi(\mathbf{d}^k; \mathbf{v}^k)$ with $\mathbf{v}^k$). To utilize the second-order information, we may perform the optimization along the direction $\mathbf{d}^k = H^k \mathbf{v}^k$, where $H^k$ is a positive definite matrix containing the second-order information. However, $\mathbf{d}^k$ is not necessarily a descent direction. To address this issue, we use the following slightly modified direction $\mathbf{p}^k$:

$$\mathbf{p}^k = \pi(\mathbf{d}^k; \mathbf{v}^k).$$

We can show that $\mathbf{p}^k$ is a descent direction (proof is provided in Supplement D). Thus, we can perform the optimization along the direction $\mathbf{p}^k$. Recall that we need to keep the current iterate in the same orthant of the previous iterate. So the following iterative scheme is proposed:

$$\mathbf{x}^k(\alpha) = \pi(\mathbf{x}^k + \alpha \mathbf{p}^k; \boldsymbol{\xi}^k), \tag{2}$$

where

$$\xi_i^k = \begin{cases} \sigma(x_i^k), & \text{if } x_i^k \neq 0, \\ \sigma(v_i^k), & \text{if } x_i^k = 0, \end{cases} \tag{3}$$

and $\alpha$ is a step size chosen by the following line search procedure: for constants $\alpha_0 > 0, \beta, \gamma \in (0,1)$ and $m = 0, 1, \cdots$, find the smallest integer $m$ with $\alpha = \alpha_0 \beta^m$ such that the following inequality holds:

$$f(\mathbf{x}^k(\alpha)) \leq f(\mathbf{x}^k) - \gamma\alpha(\mathbf{v}^k)^T\mathbf{d}^k. \tag{4}$$

However, only using the above iterative scheme may not guarantee the convergence. The main challenge is: if there exists a subsequence $\mathcal{K}$ such that $\{x_i^k\}_{\mathcal{K}}$ converges to zero, it is possible that for sufficiently large $k \in \mathcal{K}$, $|x_i^k|$ is arbitrarily small but never equal to zero (refer to the proof of Theorem 1 for more details). To address this issue, we propose a hybrid optimization scheme. Specifically, for a small constant $\epsilon > 0$, if $\mathcal{I}^k = \{i \in \{1, \cdots, n\} : 0 < |x_i^k| \leq \min(\|\mathbf{v}^k\|, \epsilon), x_i^k v_i^k < 0\}$ is not empty, we switch the iteration to the following gradient descent step (GD-step):

$$\mathbf{x}^k(\alpha) = \arg\min_{\mathbf{x}} \left\{ \nabla l(\mathbf{x}^k)^T(\mathbf{x} - \mathbf{x}^k) + \frac{1}{2\alpha}\|\mathbf{x} - \mathbf{x}^k\|^2 + r(\mathbf{x}) \right\},$$

where $\alpha$ is a step size chosen by the following line search procedure: for constants $\alpha_0 > 0, \beta, \gamma \in (0,1)$ and $m = 0, 1, \cdots$, find the smallest integer $m$ with $\alpha = \alpha_0 \beta^m$ such that the following inequality holds:

$$f(\mathbf{x}^k(\alpha)) \leq f(\mathbf{x}^k) - \frac{\gamma}{2\alpha}\|\mathbf{x}^k(\alpha) - \mathbf{x}^k\|^2. \tag{5}$$

The detailed steps of the algorithm are presented in Algorithm 1.

**Remark 3** *Algorithm 1 is similar to OWL-QN-type algorithms in [1, 3, 4, 17, 13]. However, HONOR is significantly different from them: (1) The OWL-QN-type algorithms can only handle $\ell_1$-regularized convex problems while HONOR is applicable to a class of non-convex problems beyond $\ell_1$-regularized ones. (2) The convergence analyses of the OWL-QN-type algorithms heavily rely on the convexity of the $\ell_1$-regularized problem. In contrast, the convergence analysis for HONOR is applicable to non-convex cases beyond the convex ones, which is a non-trivial extension.*

---

**Algorithm 1:** HONOR: Hybrid Optimization for NOn-convex Regularized problems
---
1 Initialize $\mathbf{x}^0$, $H^0$ and choose $\beta, \gamma \in (0,1), \epsilon > 0, \alpha_0 > 0$;
2 **for** $k = 0$ *to maxiter* **do**
3      Compute $\mathbf{v}^k \leftarrow -\diamond f(\mathbf{x}^k)$ and $\mathcal{I}^k = \{i \in \{1, \cdots, n\} : 0 < |x_i^k| \leq \epsilon^k, x_i^k v_i^k < 0\}$, where $\epsilon^k = \min(\|\mathbf{v}^k\|, \epsilon)$;
4      Initialize $\alpha \leftarrow \alpha_0$;
5      **if** $\mathcal{I}^k = \emptyset$ **then**
6          (QN-step)
7          Compute $\mathbf{d}^k \leftarrow H^k\mathbf{v}^k$ with a positive definite matrix $H^k$ using L-BFGS;
8          Alignment: $\mathbf{p}^k \leftarrow \pi(\mathbf{d}^k; \mathbf{v}^k)$;
9          **while** *Eq. (4) is not satisfied* **do**
10             $\alpha \leftarrow \alpha\beta$; $\mathbf{x}^k(\alpha) \leftarrow \pi(\mathbf{x}^k + \alpha\mathbf{p}^k; \boldsymbol{\xi}^k)$;
11          **end**
12      **else**
13          (GD-step)
14          **while** *Eq. (5) is not satisfied* **do**
15             $\alpha \leftarrow \alpha\beta$;
16             $\mathbf{x}^k(\alpha) \leftarrow \arg\min_{\mathbf{x}} \left\{\nabla l(\mathbf{x}^k)^T(\mathbf{x} - \mathbf{x}^k) + \frac{1}{2\alpha}\|\mathbf{x} - \mathbf{x}^k\|^2 + r(\mathbf{x})\right\}$;
17          **end**
18      **end**
19      $\mathbf{x}^{k+1} \leftarrow \mathbf{x}^k(\alpha)$;
20      **if** *some stopping criterion is satisfied* **then**
21          stop and return $\mathbf{x}^{k+1}$;
22      **end**
23 **end**

---

# 4 Convergence Analysis

We first present a few basic propositions and then provide the convergence theorem based on the propositions; all proofs of the presented propositions are carefully handled due to the lack of convexity. First of all, an optimality condition is presented (proof is provided in Supplement B), which will be directly used in the proof of Theorem 1.

**Proposition 1** *Let* $\bar{\mathbf{x}} = \lim_{k \in \mathcal{K}, k \to \infty} \mathbf{x}^k$, $\mathbf{v}^k = - \diamond f(\mathbf{x}^k)$ *and* $\bar{\mathbf{v}} = - \diamond f(\bar{\mathbf{x}})$, *where* $\mathcal{K}$ *is a subsequence of* $\{1, 2, \cdots, k, k+1, \cdots\}$. *If* $\liminf_{k \in \mathcal{K}, k \to \infty} |v_i^k| = 0$ *for all* $i \in \{1, \cdots, n\}$, *then* $\bar{\mathbf{v}} = \mathbf{0}$ *and* $\bar{\mathbf{x}}$ *is a Clarke critical point of problem (1).*

We subsequently show that we have a Lipschitz-continuous-like inequality in the following proposition (proof is provided in Supplement C), which is crucial to prove the final convergence theorem.

**Proposition 2** *Let* $\mathbf{v}^k = -\diamond f(\mathbf{x}^k)$, $\mathbf{x}^k(\alpha) = \pi(\mathbf{x}^k + \alpha \mathbf{p}^k; \boldsymbol{\xi}^k)$ *and* $\mathbf{q}_\alpha^k = \frac{1}{\alpha}(\pi(\mathbf{x}^k + \alpha \mathbf{p}^k; \boldsymbol{\xi}^k) - \mathbf{x}^k)$ *with* $\alpha > 0$. *Then under assumptions (A1) and (A2), we have*

$$(i) \; \nabla l(\mathbf{x}^k)^T (\mathbf{x}^k(\alpha) - \mathbf{x}^k) + r(\mathbf{x}^k(\alpha)) - r(\mathbf{x}^k) \leq -(\mathbf{v}^k)^T (\mathbf{x}^k(\alpha) - \mathbf{x}^k), \tag{6}$$

$$(ii) \; f(\mathbf{x}^k(\alpha)) \leq f(\mathbf{x}^k) - \alpha(\mathbf{v}^k)^T \mathbf{q}_\alpha^k + \frac{\alpha^2 L}{2} \|\mathbf{q}_\alpha^k\|^2. \tag{7}$$

We next show that both line search criteria in the QN-step [Eq. (4)] and the GD-step [Eq. (5)] at any iteration $k$ is satisfied in a finite number of trials (proof is provided in Supplement D).

**Proposition 3** *At any iteration* $k$ *of the HONOR algorithm, if* $\mathbf{x}^k$ *is not a Clarke critical point of problem (1), then (a) for the QN-step, there exists an* $\alpha \in [\bar{\alpha}^k, \alpha_0]$ *with* $0 < \bar{\alpha}^k \leq \alpha_0$ *such that the line search criterion in Eq. (4) is satisfied; (b) for the GD-step, the line search criterion in Eq. (5) is satisfied whenever* $\alpha \geq \beta \min(\alpha_0, (1 - \gamma)/L)$. *That is, both line search criteria at any iteration* $k$ *are satisfied in a finite number of trials.*

We are now ready to provide the convergence proof for the HONOR algorithm:

**Theorem 1** *The sequence* $\{\mathbf{x}^k\}$ *generated by the HONOR algorithm has at least a limit point and every limit point of* $\{\mathbf{x}^k\}$ *is a Clarke critical point of problem (1).*

**Proof** It follows from Proposition 3 that both line search criteria in the QN-step [Eq. (4)] and the GD-step [Eq. (5)] at each iteration can be satisfied in a finite number of trials. Let $\alpha^k$ be the accepted step size at iteration $k$. Then we have

$$f(\mathbf{x}^k) - f(\mathbf{x}^{k+1}) \geq \gamma \alpha^k (\mathbf{v}^k)^T \mathbf{d}^k = \gamma \alpha^k (\mathbf{v}^k)^T H^k \mathbf{v}^k \text{ (QN-step)}, \tag{8}$$

$$\text{or } f(\mathbf{x}^k) - f(\mathbf{x}^{k+1}) \geq \frac{\gamma}{2\alpha^k} \|\mathbf{x}^{k+1} - \mathbf{x}^k\|^2 \geq \frac{\gamma}{2\alpha_0} \|\mathbf{x}^{k+1} - \mathbf{x}^k\|^2 \text{ (GD-step)}. \tag{9}$$

Recall that $H^k$ is positive definite and $\gamma > 0, \alpha^k > 0$, which together with Eqs.(8), (9) imply that $\{f(\mathbf{x}^k)\}$ is monotonically decreasing. Thus, $\{f(\mathbf{x}^k)\}$ converges to a finite value $\bar{f}$, since $f$ is bounded from below (note that $l(\mathbf{x}) > -\infty$ and $r(\mathbf{x}) \geq 0$ for all $\mathbf{x} \in \mathbb{R}^n$). Due to the boundedness of $\{\mathbf{x}^k\}$ (see Proposition 7 in Supplement F), the sequence $\{\mathbf{x}^k\}$ generated by the HONOR algorithm has at least a limit point $\bar{\mathbf{x}}$. Since $f$ is continuous, there exists a subsequence $\mathcal{K}$ of $\{1, 2 \cdots, k, k + 1, \cdots\}$ such that

$$\lim_{k \in \mathcal{K}, k \to \infty} \mathbf{x}^k = \bar{\mathbf{x}}, \tag{10}$$

$$\lim_{k \to \infty} f(\mathbf{x}^k) = \lim_{k \in \mathcal{K}, k \to \infty} f(\mathbf{x}^k) = \bar{f} = f(\bar{\mathbf{x}}). \tag{11}$$

In the following, we prove the theorem by contradiction. Assume that $\bar{\mathbf{x}}$ is not a Clarke critical point of problem (1). Then by Proposition 1, there exists at least one $i \in \{1, \cdots, n\}$ such that

$$\liminf_{k \in \mathcal{K}, k \to \infty} |v_i^k| > 0. \tag{12}$$

We next consider the following two cases:

(a) There exist a subsequence $\tilde{\mathcal{K}}$ of $\mathcal{K}$ and an integer $\tilde{k} > 0$ such that for all $k \in \tilde{\mathcal{K}}, k \geq \tilde{k}$, the GD-step is adopted. Then for all $k \in \tilde{\mathcal{K}}, k \geq \tilde{k}$, we have

$$\mathbf{x}^{k+1} = \arg\min_{\mathbf{x}} \left\{ \nabla l(\mathbf{x}^k)^T(\mathbf{x} - \mathbf{x}^k) + \frac{1}{2\alpha^k}\|\mathbf{x} - \mathbf{x}^k\|^2 + r(\mathbf{x}) \right\}.$$

Thus, by the optimality condition of the above problem and properties of the Clarke subdifferential (Proposition 4 in Supplement A), we have

$$\mathbf{0} \in \nabla l(\mathbf{x}^k) + \frac{1}{\alpha^k}(\mathbf{x}^{k+1} - \mathbf{x}^k) + \partial^o r(\mathbf{x}^{k+1}). \tag{13}$$

Taking limits with $k \in \tilde{\mathcal{K}}$ for Eq. (9) and considering Eqs. (10), (11), we have

$$\lim_{k \in \tilde{\mathcal{K}}, k \to \infty} \|\mathbf{x}^{k+1} - \mathbf{x}^k\|^2 \leq 0 \Rightarrow \lim_{k \in \tilde{\mathcal{K}}, k \to \infty} \mathbf{x}^k = \lim_{k \in \tilde{\mathcal{K}}, k \to \infty} \mathbf{x}^{k+1} = \bar{\mathbf{x}}. \tag{14}$$

Taking limits with $k \in \tilde{\mathcal{K}}$ for Eq. (13) and considering Eq. (14), $\alpha^k \geq \beta \min(\alpha_0, (1 - \gamma)/L)$ [Proposition 3] and $\partial^o r(\cdot)$ is upper-semicontinuous (upper-hemicontinuous) [8] (see Proposition 4 in the Supplement A), we have

$$\mathbf{0} \in \nabla l(\bar{\mathbf{x}}) + \partial^o r(\bar{\mathbf{x}}) = \partial^o f(\bar{\mathbf{x}}),$$

which contradicts the assumption that $\bar{\mathbf{x}}$ is not a Clarke critical point of problem (1).

(b) There exists an integer $\hat{k} > 0$ such that for all $k \in \mathcal{K}, k \geq \hat{k}$, the QN-step is adopted. According to Remark 7 (in Supplement F), we know that the smallest eigenvalue of $H^k$ is uniformly bounded from below by a positive constant, which together with Eq. (12) implies

$$\liminf_{k \in \mathcal{K}, k \to \infty} (\mathbf{v}^k)^T H^k \mathbf{v}^k > 0. \tag{15}$$

Taking limits with $k \in \mathcal{K}$ for Eq. (8), we have

$$\lim_{k \in \mathcal{K}, k \to \infty} \gamma \alpha^k (\mathbf{v}^k)^T H^k \mathbf{v}^k \leq 0,$$

which together with $\gamma \in (0, 1), \alpha^k \in (0, \alpha_0]$ and Eq. (15) implies that

$$\lim_{k \in \mathcal{K}, k \to \infty} \alpha^k = 0. \tag{16}$$

Eq. (12) implies that there exist an integer $\check{k} > 0$ and a constant $\bar{\epsilon} > 0$ such that $\epsilon^k = \min(\|\mathbf{v}^k\|, \epsilon) \geq \bar{\epsilon}$ for all $k \in \mathcal{K}, k \geq \check{k}$. Notice that for all $k \in \mathcal{K}, k \geq \hat{k}$, the QN-step is adopted. Thus, we obtain that $\mathcal{I}^k = \{i \in \{1, \cdots, n\} : 0 < |x_i^k| \leq \epsilon^k, x_i^k v_i^k < 0\} = \emptyset$ for all $k \in \mathcal{K}, k \geq \hat{k}$. We also notice that, if $|x_i^k| \geq \bar{\epsilon}$, then there exists a constant $\bar{\alpha}_i > 0$ such that $x_i^k(\alpha) = \pi_i(x_i^k + \alpha p_i^k; \xi_i^k) = x_i^k + \alpha p_i^k$ for all $\alpha \in (0, \bar{\alpha}_i]$, as $\{p_i^k\}$ is bounded (Proposition 8 in Supplement F). Therefore, we conclude that, for all $k \in \mathcal{K}, k \geq \bar{k} = \max(\check{k}, \hat{k})$ and for all $i \in \{1, \cdots, n\}$, at least one of the following three cases must happen:

$$x_i^k = 0 \Rightarrow x_i^k(\alpha) = \pi_i(x_i^k + \alpha p_i^k; \xi_i^k) = x_i^k + \alpha p_i^k, \forall \alpha > 0,$$
$$\text{or } |x_i^k| > \epsilon^k \geq \bar{\epsilon} \Rightarrow x_i^k(\alpha) = \pi_i(x_i^k + \alpha p_i^k; \xi_i^k) = x_i^k + \alpha p_i^k, \forall \alpha \in (0, \bar{\alpha}_i],$$
$$\text{or } x_i^k v_i^k \geq 0 \Rightarrow x_i^k p_i^k \geq 0 \Rightarrow x_i^k(\alpha) = \pi_i(x_i^k + \alpha p_i^k; \xi_i^k) = x_i^k + \alpha p_i^k, \forall \alpha > 0.$$

It follows that there exists a constant $\bar{\alpha} > 0$ such that

$$\mathbf{q}_\alpha^k = \frac{1}{\alpha}(\mathbf{x}^k(\alpha) - \mathbf{x}^k) = \mathbf{p}^k, \forall k \in \mathcal{K}, k \geq \bar{k}, \alpha \in (0, \bar{\alpha}]. \tag{17}$$

Thus, considering $|p_i^k| = |\pi_i(d_i^k; v_i^k)| \leq |d_i^k|$ and $v_i^k p_i^k \geq v_i^k d_i^k$ for all $i \in \{1, \cdots, n\}$, we have

$$\|\mathbf{q}_\alpha^k\|^2 = \|\mathbf{p}^k\|^2 \leq \|\mathbf{d}^k\|^2 = (\mathbf{v}^k)^T (H^k)^2 \mathbf{v}^k, \forall k \in \mathcal{K}, k \geq \bar{k}, \alpha \in (0, \bar{\alpha}], \tag{18}$$
$$(\mathbf{v}^k)^T \mathbf{q}_\alpha^k = (\mathbf{v}^k)^T \mathbf{p}^k \geq (\mathbf{v}^k)^T \mathbf{d}^k = (\mathbf{v}^k)^T H^k \mathbf{v}^k, \forall k \in \mathcal{K}, k \geq \bar{k}, \alpha \in (0, \bar{\alpha}]. \tag{19}$$

According to Proposition 8 (in Supplement F), we know that the largest eigenvalue of $H^k$ is uniformly bounded from above by some positive constant $M$. Thus, we have

$$(\mathbf{v}^k)^T (H^k)^2 \mathbf{v}^k \leq \frac{2}{\alpha L}(\mathbf{v}^k)^T H^k \mathbf{v}^k - \left(\frac{2}{\alpha L} - M\right)(\mathbf{v}^k)^T H^k \mathbf{v}^k, \forall k,$$

which together with Eqs. (18), (19) and $\mathbf{d}^k = H^k \mathbf{v}^k$ implies

$$\|\mathbf{q}_\alpha^k\|^2 \leq \frac{2}{\alpha L}(\mathbf{v}^k)^T \mathbf{q}_\alpha^k - \left(\frac{2}{\alpha L} - M\right)(\mathbf{v}^k)^T \mathbf{d}^k, \forall k \in \mathcal{K}, k \geq \bar{k}, \alpha \in (0, \bar{\alpha}]. \qquad (20)$$

Considering Eqs. (7), (20), we have

$$f(\mathbf{x}^k(\alpha)) \leq f(\mathbf{x}^k) - \alpha\left(1 - \frac{\alpha L M}{2}\right)(\mathbf{v}^k)^T \mathbf{d}^k, \forall k \in \mathcal{K}, k \geq \bar{k}, \alpha \in (0, \bar{\alpha}],$$

which together with $(\mathbf{v}^k)^T \mathbf{d}^k = (\mathbf{v}^k)^T H^k \mathbf{v}^k \geq 0$ implies that the line search criterion in the QN-step [Eq. (4)] is satisfied if

$$1 - \frac{\alpha L M}{2} \geq \gamma \,, 0 < \alpha \leq \alpha_0 \text{ and } 0 < \alpha \leq \bar{\alpha}, \forall k \in \mathcal{K}, k \geq \bar{k}.$$

Considering the backtracking form of the line search in QN-step [Eq. (4)], we conclude that the line search criterion in the QN-step [Eq. (4)] is satisfied whenever

$$\alpha^k \geq \beta \min(\min(\bar{\alpha}, \alpha_0), 2(1-\gamma)/(LM)) > 0, \forall k \in \mathcal{K}, k \geq \bar{k}.$$

This leads to a contradiction with Eq. (16).

By (a) and (b), we conclude that $\bar{\mathbf{x}} = \lim_{k \in \mathcal{K}, k \to \infty} \mathbf{x}^k$ is a Clarke critical point of problem (1). $\qquad \square$

## 5 Experiments

In this section, we evaluate the efficiency of HONOR on solving the non-convex regularized logistic regression problem[1] by setting $l(\mathbf{x}) = 1/N \sum_{i=1}^{N} \log(1 + \exp(-y_i \mathbf{a}_i^T \mathbf{x}))$, where $\mathbf{a}_i \in \mathbb{R}^n$ is the $i$-th sample associated with the label $y_i \in \{1, -1\}$. Three non-convex regularizers (LSP, MCP and SCAD) are included in experiments, where the parameters are set as $\lambda = 1/N$ and $\theta = 10^{-2}\lambda$ ($\theta$ is set as $2 + 10^{-2}\lambda$ for SCAD as it requires $\theta > 2$). We compare HONOR with the non-convex solver[2] GIST [14] on three large-scale, high-dimensional and sparse data sets which are summarized in Table 1. All data sets can be downloaded from http://www.csie.ntu.edu.tw/~cjlin/libsvmtools/datasets/.

All algorithms are implemented in Matlab 2015a under a Linux operating system and executed on an Intel Core i7-4790 CPU (@3.6GHz) with 32GB memory. We choose the starting points

Table 1: Data set statistics.

| datasets | kdd2010a | kdd2010b | url |
|---|---|---|---|
| ♯ samples $N$ | 510,302 | 748,401 | 2,396,130 |
| dimensionality $n$ | 20,216,830 | 29,890,095 | 3,231,961 |

$\mathbf{x}^0$ for the compared algorithms using the same random vector whose entries are i.i.d. sampled from the standard Gaussian distribution. We terminate the compared algorithms if the relative change of two consecutive objective function values is less than $10^{-5}$ or the number of iterations exceeds 1000 (HONOR) or 10000 (GIST). For HONOR, we set $\gamma = 10^{-5}, \beta = 0.5, \alpha_0 = 1$ and the number of unrolling steps in L-BFGS as $m = 10$. For GIST, we use the non-monotone line search in experiments as it usually performs better than its monotone counterpart. To show how the convergence behavior of HONOR varies over the parameter $\epsilon$, we use three values: $\epsilon = 10^{-10}, 10^{-6}, 10^{-2}$.

We report the objective function value (in log-scale) vs. CPU time (in seconds) plots in Figure 1. We can observe from Figure 1 that: (1) If $\epsilon$ is set to a small value, the QN-step is adopted at almost all steps in HONOR and HONOR converges significantly faster than GIST for all three non-convex

regularizers on all three data sets. This shows that using the second-order information greatly speeds up the convergence. (2) When $\epsilon$ increases, the ratio of the GD-step adopted in HONOR increases. Meanwhile, the convergence performance of HONOR generally degrades. In some cases, setting a slightly larger $\epsilon$ and adopting a small number of GD steps even sligtly boosts the convergence performance of HONOR (the green curves in the first row). But setting $\epsilon$ to a very small value is always safe to guarantee the fast convergence of HONOR. (3) When $\epsilon$ is large enough, the GD steps dominate all iterations of HONOR and HONOR converge much slower. In this case, HONOR converges even slower than GIST. The reason is that, at each iteration of HONOR, extra computational cost is required in addition to the basic computation in the GD-step. Moreover, the non-monotone line search is used in GIST while the monotone line search is adopted in the GD-step. (4) In some cases (the first row), GIST is trapped in a local solution which has a much larger objective function value than HONOR with a small $\epsilon$. This implies that HONOR may have a potential of escaping from high error plateau which often exists in high dimensional non-convex problems. These results show the great potential of HONOR for solving large-scale non-convex sparse learning problems.

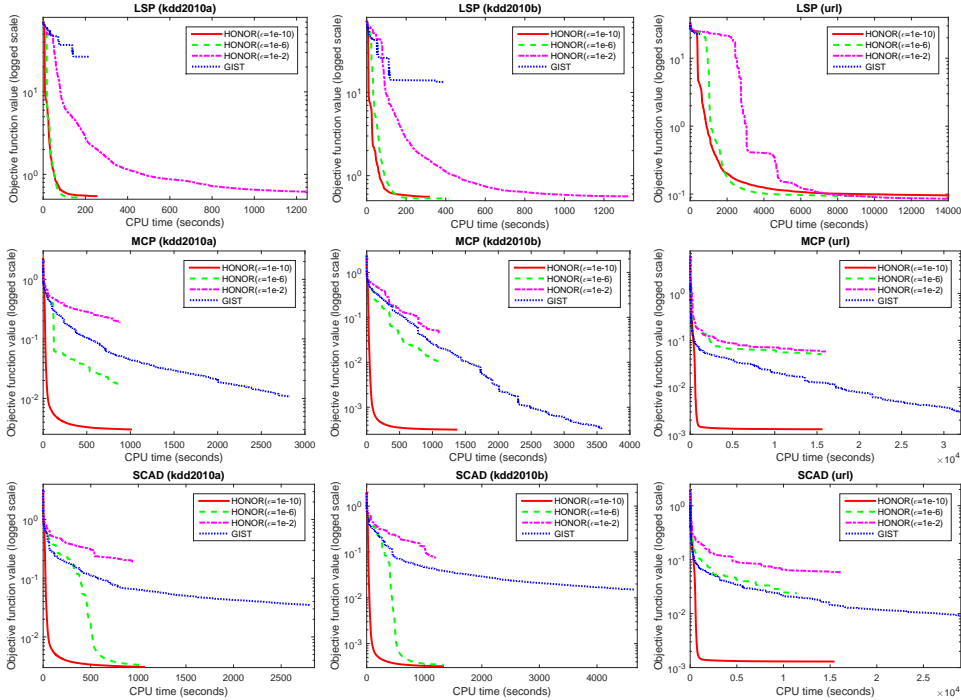

Figure 1: Objective function value (in log-scale) vs. CPU time (in seconds) plots for different non-convex regularizers and different large-scale and high-dimensional data sets. The ratios of the GD-step adopted in HONOR are: LSP (kdd2010a): $0\%, 1\%, 34\%$; LSP (kdd2010b): $0\%, 2\%, 27\%$; LSP (url): $0.1\%, 2\%, 35\%$; MCP (kdd2010a): $0\%, 88\%, 100\%$; MCP (kdd2010b): $0\%, 89\%, 100\%$; MCP (url): $0\%, 97\%, 100\%$; SCAD (kdd2010a): $0\%, 43\%, 100\%$; SCAD (2010b): $0\%, 32\%, 99.5\%$; SCAD (url): $0\%, 79\%, 100\%$.

## 6   Conclusions

In this paper, we propose an efficient optimization algorithm called HONOR for solving non-convex regularized sparse learning problems. HONOR incorporates the second-order information to speed up the convergence in practice and uses a carefully designed hybrid optimization scheme to guarantee the convergence in theory. Experiments are conducted on large-scale data sets and results show that HONOR converges significantly faster than state-of-the-art algorithms. In our future work, we plan to develop parallel/distributed variants of HONOR to tackle much larger data sets.

## Acknowledgements

This work is supported in part by research grants from NIH (R01 LM010730, U54 EB020403) and NSF (IIS- 0953662, III-1539991, III-1539722).

## Footnotes

[1]We do not include the term $\delta\|\mathbf{x}\|^2$ in the objective and find that the proposed algorithm still works well.

[2]We do not involve SparseNet, DC programming and DC-PN in comparison, because (1) adapting SparseNet to the logistic regression problem is challenging; (2) DC programming is shown to be much inferior to GIST; (3) The objective function value of DC-PN is larger than GIST in most cases [18].

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
