[Supplementary Material]

# Supplementary Material for "HONOR: Hybrid Optimization for NOn-convex Regularized problems"

## A  Properties of Clarke Subdifferential

**Proposition 4** *According to [8], we have the following properties for Clarke Subdifferential:*

(1) *If $f(\mathbf{x})$ is continuously differentiable then $\partial^o f(\mathbf{x}) = \{\nabla f(\mathbf{x})\}$.*

(2) *Let $f(\mathbf{x})$ and $g(\mathbf{x})$ be locally Lipschitz continuous on $\mathbb{R}^n$, Then for any $\mathbf{x} \in \mathbb{R}^n$, we have*

$$\partial^o(f(\mathbf{x}) + g(\mathbf{x})) \subseteq \partial^o f(\mathbf{x}) + \partial^o g(\mathbf{x}).$$

*If one of them is continuously differentiable then equality holds.*

(3) *We have the following equivalent way to express $\partial^o f(\bar{\mathbf{x}})$:*

$$\partial^o f(\bar{\mathbf{x}}) = co\left\{ \mathbf{g} \in \mathbb{R}^n : \mathbf{g} = \lim_{k \to \infty} \nabla f(\mathbf{x}^k), \mathbf{x}^k \to \bar{\mathbf{x}}, \mathbf{x}^k \in \mathcal{D} \right\},$$

*where the set $\mathcal{D}$ is the set of points over which $f$ is differentiable; co denotes the convex hull of a set.*

(4) *For a locally Lipschitz continuous function $f(\mathbf{x})$, $\partial^o f(\mathbf{x})$ is nonempty, convex and compact for each $\mathbf{x} \in \mathbb{R}^n$. As a set-valued map $\partial^o f(\mathbf{x})$ is locally bounded and has a closed graph and hence is upper-semicontinuous (upper-hemicontinuous), that is: for every sequence $\{\mathbf{x}^k\} \to \mathbf{x}$ and every sequence $\{\mathbf{y}^k\} \to \mathbf{y}$ with $\mathbf{y}^k \in \partial^o f(\mathbf{x}^k)$, we have $\mathbf{y} \in \partial^o f(\mathbf{x})$.*

**Remark 4** *For the property (2) above, the equality holds for the subdifferential of convex functions without requiring that one of them is continuously differentiable.*

It is generally difficult to compute the Clarke subdifferential of a non-convex function based on its definition. However, according to the above properties and the special structure of the non-convex regularizer, we can obtain the the Clarke subdifferential of $f(\mathbf{x})$ in problem (1) in the following proposition:

**Proposition 5** *Let $f(\mathbf{x}) = l(\mathbf{x}) + r(\mathbf{x})$ and $\mathbf{g} \in \partial^o f(\mathbf{x})$. Then, under assumptions (A1) and (A2), the $i$-th entry of $\mathbf{g}$ is*

$$\begin{aligned}
g_i &= \nabla_i l(\mathbf{x}) + \rho'(|x_i|), & \text{if } x_i > 0, \\
g_i &= \nabla_i l(\mathbf{x}) - \rho'(|x_i|), & \text{if } x_i < 0, \\
g_i &\in [\nabla_i l(\mathbf{x}) - \rho'(0), \nabla_i l(\mathbf{x}) + \rho'(0)], & \text{if } x_i = 0.
\end{aligned}$$

**Proof** According to the properties (1) and (2) in Proposition 4, we obtain that, if $x_i \neq 0$, then $g_i = \nabla_i l(\mathbf{x}) + \sigma(x_i)\rho'(|x_i|)$, which immediately implies the first two. Further considering the property (3) in Proposition 4, the last one is easily obtained. $\square$

## B  Proof of Proposition 1

**Proof** We firstly use contradiction to prove that if $\liminf_{k \in \mathcal{K}, k \to \infty} |v_i^k| = 0$ for all $i \in \{1, \cdots, n\}$, then $\bar{\mathbf{v}} = \mathbf{0}$. Assume that $\liminf_{k \in \mathcal{K}, k \to \infty} |v_i^k| = 0$ for all $i \in \{1, \cdots, n\}$ but $\bar{\mathbf{v}} \neq \mathbf{0}$. Then there exists at least one $i \in \{1, \cdots, n\}$ such that $\bar{v}_i = - \diamond_i f(\bar{\mathbf{x}}) \neq 0$. We consider the following two cases:

(1) If $\bar{x}_i \neq 0$, then we have $\liminf_{k \in \mathcal{K}, k \to \infty} |v_i^k| = |\bar{v}_i| \neq 0$, leading to a contradiction with that $\liminf_{k \in \mathcal{K}, k \to \infty} |v_i^k| = 0$ for all $i \in \{1, \cdots, n\}$.

(2) If $\bar{x}_i = 0$, then $\bar{v}_i = - \diamond_i f(\bar{\mathbf{x}}) \neq 0$ implies that

$$\nabla_i l(\bar{\mathbf{x}}) + \rho'(0) > \nabla_i l(\bar{\mathbf{x}}) - \rho'(0) > 0, \text{ or } \nabla_i l(\bar{\mathbf{x}}) - \rho'(0) < \nabla_i l(\bar{\mathbf{x}}) + \rho'(0) < 0. \tag{21}$$

By the definition of $v_i^k = -\diamond_i f(\mathbf{x}^k)$, we know that

$$-(\nabla_i l(\mathbf{x}^k) + \rho'(0)) \le v_i^k \le -(\nabla_i l(\mathbf{x}^k) - \rho'(0)).$$

Taking limits of the above inequalities, we have

$$-(\nabla_i l(\bar{\mathbf{x}}) + \rho'(0)) \le \liminf_{k \in \mathcal{K}, k \to \infty} v_i^k \le -(\nabla_i l(\bar{\mathbf{x}}) - \rho'(0)), \text{ and}$$

$$-(\nabla_i l(\bar{\mathbf{x}}) + \rho'(0)) \le \limsup_{k \in \mathcal{K}, k \to \infty} v_i^k \le -(\nabla_i l(\bar{\mathbf{x}}) - \rho'(0)),$$

which together with Eq. (21) imply that

$$\liminf_{k \in \mathcal{K}, k \to \infty} |v_i^k| \ne 0.$$

This leads to a contradiction with that $\liminf_{k \in \mathcal{K}, k \to \infty} |v_i^k| = 0$ for all $i \in \{1, \cdots, n\}$. Therefore, if $\liminf_{k \in \mathcal{K}, k \to \infty} |v_i^k| = 0$ for all $i \in \{1, \cdots, n\}$, then $\bar{\mathbf{v}} = \mathbf{0}$.

To complete the proof, we next prove that $\bar{\mathbf{x}}$ is a Clarke critical point of problem (1) if $\bar{\mathbf{v}} = \mathbf{0}$. According to the definition of the pseudo gradient and Proposition 5, it is easy to verify that

$$\diamond f(\bar{\mathbf{x}}) = \underset{\bar{\mathbf{g}} \in \partial^o f(\bar{\mathbf{x}})}{\arg\min} \|\bar{\mathbf{g}}\|. \tag{22}$$

Thus, $\mathbf{0} \in \partial^o f(\bar{\mathbf{x}}) \Leftrightarrow \diamond f(\bar{\mathbf{x}}) = \mathbf{0} \Leftrightarrow \bar{\mathbf{v}} = \mathbf{0}$ and hence $\bar{\mathbf{x}}$ is a Clarke critical point of problem (1) if $\bar{\mathbf{v}} = \mathbf{0}$. $\qquad\square$

## C   Proof of Proposition 2

**Proof** $(i)$ Based on the definition of $\mathbf{x}^k(\alpha)$, we know that $x_i^k(\alpha) x_i^k \ge 0$. We next prove for all $i \in \{1, \cdots, n\}$, the following inequality holds by considering two cases:

$$\nabla_i l(\mathbf{x}^k)(x_i^k(\alpha) - x_i^k) + \rho(|x_i^k(\alpha)|) - \rho(|x_i^k|) \le -v_i^k(x_i^k(\alpha) - x_i^k). \tag{23}$$

(a) If $x_i^k \ne 0$, then $x_i^k(\alpha) x_i^k \ge 0$ implies $|x_i^k(\alpha)| - |x_i^k| = \sigma(x_i^k)(x_i^k(\alpha) - x_i^k)$. By the concavity of $\rho(\cdot)$, we have $\rho(|x_i^k(\alpha)|) - \rho(|x_i^k|) \le \rho'(|x_i^k|)(|x_i^k(\alpha)| - |x_i^k|) = \rho'(|x_i^k|)\sigma(x_i^k)(x_i^k(\alpha) - x_i^k)$, which together with $\nabla_i l(\mathbf{x}^k) + \rho'(|x_i^k|)\sigma(x_i^k) = -v_i^k$ (by noticing that $x_i^k \ne 0$) implies that Eq. (23) holds.

(b) If $x_i^k = 0$, then we have $x_i^k(\alpha) = \pi_i(\alpha p_i^k; \sigma(v_i^k)) = \alpha p_i^k$. We next focus on the case (b) in the following two subcases:

(1) If $p_i^k \ne 0$, then $|x_i^k(\alpha)| = \alpha\sigma(p_i^k)p_i^k = \sigma(v_i^k)(\alpha p_i^k) = \sigma(v_i^k)x_i^k(\alpha)$, which together with the concavity of $\rho(\cdot)$ and $\rho(x_i^k) = \rho(0) = 0$ implies that $\nabla_i l(\mathbf{x}^k)(x_i^k(\alpha) - x_i^k) + \rho(|x_i^k(\alpha)|) - \rho(|x_i^k|) \le \nabla_i l(\mathbf{x}^k)x_i^k(\alpha) + \rho'(0)\sigma(v_i^k)x_i^k(\alpha)$, which together with $x_i^k = 0, v_i^k \ne 0$ and $\nabla_i l(\mathbf{x}^k) + \rho'(0)\sigma(v_i^k) = -v_i^k$ whenever $x_i^k = 0$ and $v_i^k \ne 0$ implies that Eq. (23) holds.

(2) If $p_i^k = 0$, then $x_i^k(\alpha) = x_i^k = 0$, which together with the fact that $\rho(0) = 0$ implies Eq. (23) holds.

Combining (a) and (b), we obtain that Eq. (23) holds for all $i \in \{1, \cdots, n\}$, which together with the definition of $r(\cdot)$ in the assumption (A2) implies that Eq. (6) holds.

$(ii)$ Since $\nabla l(\mathbf{x})$ is Lipschitz continuous with constant $L$, we have

$$l(\mathbf{x}^k(\alpha)) \le l(\mathbf{x}^k) + \nabla l(\mathbf{x}^k)^T(\mathbf{x}^k(\alpha) - \mathbf{x}^k) + \frac{L}{2}\|\mathbf{x}^k(\alpha) - \mathbf{x}^k\|^2.$$

It follows that

$$f(\mathbf{x}^k(\alpha)) \le f(\mathbf{x}^k) + \nabla l(\mathbf{x}^k)^T(\mathbf{x}^k(\alpha) - \mathbf{x}^k) + r(\mathbf{x}^k(\alpha)) - r(\mathbf{x}^k) + \frac{L}{2}\|\mathbf{x}^k(\alpha) - \mathbf{x}^k\|^2,$$

which together with Eq. (6) and $\mathbf{q}_\alpha^k = \frac{1}{\alpha}(\mathbf{x}^k(\alpha) - \mathbf{x}^k)$ implies that Eq. (7) holds. $\qquad\square$

# D   Proof of Proposition 3 and Auxiliary Propositions

We present the following proposition which is useful to prove Proposition 3.

**Proposition 6** *Let $f(\mathbf{x}) = l(\mathbf{x}) + r(\mathbf{x})$ and assumptions (A1) and (A2) hold. If $\mathbf{p}^k = \pi(\mathbf{d}^k; \mathbf{v}^k)$ is a non-zero vector, where $\mathbf{v}^k = - \diamond f(\mathbf{x}^k)$, then the directional derivative of $f(\mathbf{x})$ at $\mathbf{x} = \mathbf{x}^k$ along the direction $\mathbf{p}^k$ defined as*

$$f'(\mathbf{x}^k; \mathbf{p}^k) = \lim_{\alpha \downarrow 0} \frac{f(\mathbf{x}^k + \alpha \mathbf{p}^k) - f(\mathbf{x}^k)}{\alpha} \tag{24}$$

*exists and $f'(\mathbf{x}^k; \mathbf{p}^k) = -(\mathbf{v}^k)^T \mathbf{p}^k < 0$.*

**Proof**   Recall that $l(\mathbf{x})$ is continuously differentiable based on the assumption (A1), so by the mean value theorem, for any $\alpha > 0$, there exists an $\tilde{\alpha} \in [0, \alpha]$ such that $l(\mathbf{x}^k + \alpha \mathbf{p}^k) - l(\mathbf{x}^k) = \alpha(\mathbf{p}^k)^T \nabla l(\mathbf{x}^k + \tilde{\alpha}\mathbf{p}^k)$. Thus, we have

$$\lim_{\alpha \downarrow 0} \frac{l(\mathbf{x}^k + \alpha \mathbf{p}^k) - l(\mathbf{x}^k)}{\alpha} = \lim_{\alpha \downarrow 0} \frac{\alpha(\mathbf{p}^k)^T \nabla l(\mathbf{x}^k + \tilde{\alpha}\mathbf{p}^k)}{\alpha} = \nabla l(\mathbf{x}^k)^T \mathbf{p}^k.$$

When $x_i^k \neq 0$, there exists an $\alpha_0 > 0$ such that for any $\alpha \in (0, \alpha_0]$, $\sigma(x_i^k + \alpha p_i^k) = \sigma(x_i^k) \neq 0$. Based on Remark 1, we know that $\rho(|x_i|)$ is continuously differentiable with respect to $x_i$ in $(-\infty, 0) \cup (0, \infty)$. Thus, by the mean value theorem, there exists an $\tilde{\alpha} \in [0, \alpha]$ such that $\rho(|x_i^k + \alpha p_i^k|) - \rho(|x_i^k|) = \partial \rho(|x_i^k + \tilde{\alpha}p_i^k|)/\partial(x_i^k + \tilde{\alpha}p_i^k)|x_i^k + \alpha p_i^k - x_i^k| = \rho'(|x_i^k + \tilde{\alpha}p_i^k|)\sigma(x_i^k + \tilde{\alpha}p_i^k)|\alpha p_i^k|$. Therefore, we have

$$\lim_{\alpha \downarrow 0} \frac{\rho(|x_i^k + \alpha p_i^k|) - \rho(|x_i^k|)}{\alpha} = \lim_{\alpha \downarrow 0} \rho'(|x_i^k + \tilde{\alpha}p_i^k|)\sigma(x_i^k + \tilde{\alpha}p_i^k)|p_i^k| = \rho'(|x_i^k|)\sigma(x_i^k)p_i^k.$$

When $x_i^k = 0$, by the continuous differentiability of $\rho(\cdot)$ in $[0, \infty)$ and the mean value theorem, we have for any $\alpha > 0$, there exists an $\tilde{\alpha} \in [0, \alpha]$ such that $\rho(|x_i^k + \alpha p_i^k|) - \rho(|x_i^k|) = \rho(|\alpha p_i^k|) - \rho(0) = \partial \rho(|\tilde{\alpha}p_i^k|)/\partial(|\tilde{\alpha}p_i^k|)(|\alpha p_i^k| - 0) = \rho'(|\tilde{\alpha}p_i^k|)|\alpha p_i^k|$. Thus, we have

$$\lim_{\alpha \downarrow 0} \frac{\rho(|x_i^k + \alpha p_i^k|) - \rho(|x_i^k|)}{\alpha} = \lim_{\alpha \downarrow 0} \rho'(|\tilde{\alpha}p_i^k|)|p_i^k| = \rho'(0)|p_i^k| = \rho'(0)\sigma(v_i^k)p_i^k.$$

Therefore, according to Eq. (24) and $f(\mathbf{x}) = l(\mathbf{x}) + r(\mathbf{x}) = l(\mathbf{x}) + \sum_{i=1}^n \rho(|x_i|)$, we have

$$f'(\mathbf{x}^k; \mathbf{p}^k) = \lim_{\alpha \downarrow 0} \frac{l(\mathbf{x}^k + \alpha \mathbf{p}^k) - l(\mathbf{x}^k)}{\alpha} + \sum_{i=1}^n \lim_{\alpha \downarrow 0} \frac{\rho(|x_i^k + \alpha p_i^k|) - \rho(|x_i^k|)}{\alpha}$$

$$= \nabla l(\mathbf{x}^k)^T \mathbf{p}^k + \sum_{i \in \mathcal{A}_k} \rho'(|x_i^k|)\sigma(x_i^k)p_i^k + \sum_{i \in \mathcal{A}_k^c} \rho'(0)\sigma(v_i^k)p_i^k,$$

where $\mathcal{A}_k = \{i : x_i^k \neq 0\}$, $\mathcal{A}_k^c = \{i : x_i^k = 0\}$. Rearranging the above equality, we have

$$f'(\mathbf{x}^k; \mathbf{p}^k) = \sum_{i \in \mathcal{A}_k} \left( \nabla_i l(\mathbf{x}^k) + \rho'(|x_i^k|)\sigma(x_i^k) \right) p_i^k + \sum_{i \in \mathcal{A}_k^c} \left( \nabla_i l(\mathbf{x}^k) + \rho'(0)\sigma(v_i^k) \right) p_i^k$$

$$= \sum_{i=1}^n -v_i^k p_i^k = -(\mathbf{v}^k)^T \mathbf{p}^k < 0,$$

where the second equality follows from the definition of $\diamond_i f(\mathbf{x}^k)$ and $v_i^k = - \diamond_i f(\mathbf{x}^k)$; the last inequality follows from $\mathbf{p}^k = \pi(\mathbf{d}^k; \mathbf{v}^k)$ and the condition $\mathbf{p}^k \neq \mathbf{0}$.   □

**Remark 5** *For a convex function, the directional derivative always exists. However, for a non-convex function, we are required to address the issue whether the directional derivative exists based on its definition.*

Based on Proposition 6, we prove Proposition 3 as follows:

**Proof of Proposition 3** (a) For QN-step, let's define

$$\mathcal{B}_k = \{i : x_i^k p_i^k < 0\} \text{ and } \bar{\alpha}_1^k = \begin{cases} \min_{i \in \mathcal{B}_k} \frac{|x_i^k|}{|p_i^k|}, & \text{if } \mathcal{B}_k \neq \emptyset, \\ +\infty, & \text{otherwise.} \end{cases}$$

Then for all $\alpha \in (0, \bar{\alpha}_1^k)$, we have

$$\mathbf{x}^k(\alpha) = \pi(\mathbf{x}^k + \alpha \mathbf{p}^k; \boldsymbol{\xi}^k) = \mathbf{x}^k + \alpha \mathbf{p}^k. \tag{25}$$

Define

$$s(\alpha) = f(\mathbf{x}^k + \alpha \mathbf{p}^k), \; h(\alpha) = \frac{s(\alpha) - s(0)}{\alpha}.$$

Recalling the definition of the directional derivative in Eq. (24), $\gamma \in (0, 1)$ and Proposition 6, we have

$$\lim_{\alpha \downarrow 0} \frac{s(\alpha) - s(0)}{\alpha} = -(\mathbf{v}^k)^T \mathbf{p}^k \leq -(\mathbf{v}^k)^T \mathbf{d}^k < -\gamma(\mathbf{v}^k)^T \mathbf{d}^k,$$

where the first inequality follows from $v_i^k p_i^k \geq v_i^k d_i^k$ and the last inequality follows from $\gamma \in (0, 1)$ and $(\mathbf{v}^k)^T \mathbf{d}^k = (\mathbf{v}^k)^T H^k \mathbf{v}^k > 0$ whenever $\mathbf{x}^k$ is not a Clarke critical point of problem (1). Thus, by recalling that $h(\alpha)$ is continuous in $(0, \infty)$, there exists an $\bar{\alpha}_2^k \in (0, \min(\alpha_0, \bar{\alpha}_1^k))$ such that

$$\frac{s(\alpha) - s(0)}{\alpha} \leq -\gamma(\mathbf{v}^k)^T \mathbf{d}^k, \; \forall 0 < \alpha \leq \bar{\alpha}_2^k. \tag{26}$$

Thus, considering Eq. (26) and the backtracking form of the line search in QN-step (Eq. (4)), there exists an $\alpha$ with $\alpha \geq \bar{\alpha}^k = \beta \bar{\alpha}_2^k > 0$ such that

$$\frac{s(\alpha) - s(0)}{\alpha} \leq -\gamma(\mathbf{v}^k)^T \mathbf{d}^k. \tag{27}$$

Substituting the definition of $s(\alpha)$ into Eq. (27) and considering that Eq. (25) holds for all $\alpha \in (0, \bar{\alpha}_1^k)$, we obtain that there exists an $\alpha \in [\bar{\alpha}^k, \alpha_0]$ such that the line search criterion in Eq. (4) is satisfied.

(b) For GD-step, we have

$$\nabla l(\mathbf{x}^k)^T(\mathbf{x}^k(\alpha) - \mathbf{x}^k) + \frac{1}{2\alpha}\|\mathbf{x}^k(\alpha) - \mathbf{x}^k\|^2 + r(\mathbf{x}^k(\alpha)) \leq r(\mathbf{x}^k). \tag{28}$$

Noticing that $\nabla l(\mathbf{x})$ is Lipschitz continuous with constant $L$, we have

$$l(\mathbf{x}^k(\alpha)) \leq l(\mathbf{x}^k) + \nabla l(\mathbf{x}^k)^T(\mathbf{x}^k(\alpha) - \mathbf{x}^k) + \frac{L}{2}\|\mathbf{x}^k(\alpha) - \mathbf{x}^k\|^2,$$

which together with Eq. (28) and $f(\mathbf{x}) = l(\mathbf{x}) + r(\mathbf{x})$ implies that

$$f(\mathbf{x}^k(\alpha)) \leq f(\mathbf{x}^k) - \frac{1 - \alpha L}{2\alpha}\|\mathbf{x}^k(\alpha) - \mathbf{x}^k\|^2.$$

Thus, the line search in Eq. (5) is satisfied if

$$\gamma \leq 1 - \alpha L \text{ and } 0 < \alpha \leq \alpha_0.$$

Considering the backtracking form of the line search in GD-step (Eq. (5)), we obtain that the line search criterion in Eq. (5) is satisfied whenever $\alpha \geq \beta \min(\alpha_0, (1 - \gamma)/L)$. $\qquad \square$

## E  BFGS and L-BFGS

Assume that we are given an approximate inverse Hessian matrix $H^k$ at $\mathbf{x} = \mathbf{x}^k$. BFGS updates the inverse Hessian matrix $H^{k+1}$ at $\mathbf{x} = \mathbf{x}^{k+1}$ as:

$$H^{k+1} = (V^k)^T H^k V^k + \rho^k \mathbf{s}^k (\mathbf{s}^k)^T, \tag{29}$$

where $V^k = I - \rho^k \mathbf{y}^k (\mathbf{s}^k)^T$, $\mathbf{s}^k = \mathbf{x}^{k+1} - \mathbf{x}^k$, $\mathbf{y}^k = \nabla l(\mathbf{x}^{k+1}) - \nabla l(\mathbf{x}^k)$, $\rho^k = ((\mathbf{y}^k)^T \mathbf{s}^k)^{-1}$. It is easy to verify that $H^{k+1} \succ 0$, if $H^k \succ 0$ and $\rho^k > 0$ [15].

L-BFGS [15] updates the inverse Hessian matrix by unrolling the update from BFGS back to $m$ steps:

$$
\begin{aligned}
H^k &= (V^{k-1})^T H^{k-1} V^{k-1} + \rho^{k-1} \mathbf{s}^{k-1} (\mathbf{s}^{k-1})^T \\
&= (V^{k-1})^T (V^{k-2})^T H^{k-2} V^{k-2} V^{k-1} \\
&\quad + (V^{k-1})^T \mathbf{s}^{k-2} \rho^{k-2} (\mathbf{s}^{k-2})^T V^{k-1} \\
&\quad + \rho^{k-1} \mathbf{s}^{k-1} (\mathbf{s}^{k-1})^T \\
&= \left( U^{k,m} \right)^T H^{k-m} U^{k,m} \\
&\quad + \rho^{k-m} \left( U^{k,m-1} \right)^T \mathbf{s}^{k-m} (\mathbf{s}^{k-m})^T U^{k,m-1} \\
&\quad + \rho^{k-m+1} \left( U^{k,m-2} \right)^T \mathbf{s}^{k-m+1} (\mathbf{s}^{k-m+1})^T U^{k,m-2} \\
&\quad + \cdots \\
&\quad + \rho^{k-2} (V^{k-1})^T \mathbf{s}^{k-2} (\mathbf{s}^{k-2})^T V^{k-1} \\
&\quad + \rho^{k-1} \mathbf{s}^{k-1} (\mathbf{s}^{k-1})^T,
\end{aligned}
\tag{30}
$$

where $U^{k,m} = V^{k-m} V^{k-m+1} \cdots V^{k-1}$. For the L-BFGS, we need *not* explicitly store the approximated inverse Hessian matrix. Instead, we only require matrix-vector multiplications at each iteration, which can be implemented by a two-loop recursion with a time complexity of $O(mn)$ [15]. Thus, we only store $2m$ vectors of length $n$: $\mathbf{s}^{k-1}, \mathbf{s}^{k-2}, \cdots, \mathbf{s}^{k-m}$ and $\mathbf{y}^{k-1}, \mathbf{y}^{k-2}, \cdots, \mathbf{y}^{k-m}$ with a storage complexity of $O(mn)$, which is very useful when $n$ is large. In practice, L-BFGS updates $H^{k-m}$ as $\mu^k I$, where $\mu^k = \min(10^{10}, \max(10^{-10}, (\mathbf{s}^k)^T \mathbf{y}^k / \|\mathbf{y}^k\|^2))$.

## F    Properties of L-BFGS

We first show that some key sequences are bounded, which are critical for establishing some important properties of L-BFGS.

**Proposition 7** *The sequence $\{\mathbf{x}^k\}$ generated by the HONOR algorithm is bounded. Let $\mathbf{s}^k = \mathbf{x}^{k+1} - \mathbf{x}^k$, $\mathbf{y}^k = \nabla l(\mathbf{x}^{k+1}) - \nabla l(\mathbf{x}^k)$. Then $\{\mathbf{s}^k\}$, $\{\mathbf{y}^k\}$ and $\{\mathbf{v}^k\}$ are also bounded.*

**Proof** Proposition 3 guarantees that both line search criteria in QN-step (Eq. (4)) and GD-step (Eq. (5)) can be satisfied in a finite number of trials with some $\alpha^k > 0$. Thus, we have

$$
f(\mathbf{x}^k) - f(\mathbf{x}^{k+1}) \geq \gamma \alpha^k (\mathbf{v}^k)^T \mathbf{d}^k = \gamma \alpha^k (\mathbf{v}^k)^T H^k \mathbf{v}^k \geq 0 \text{ (QN-step)},
$$
$$
\text{or } f(\mathbf{x}^k) - f(\mathbf{x}^{k+1}) \geq \frac{\gamma}{2\alpha^k} \|\mathbf{x}^{k+1} - \mathbf{x}^k\|^2 \geq 0 \text{ (GD-step)},
\tag{31}
$$

which imply that $\{f(\mathbf{x}^k)\}$ is decreasing. Hence for all $k \geq 1$, $f(\mathbf{x}^k) \leq f(\mathbf{x}^0)$. Assume that $\{\mathbf{x}^k\}$ is unbounded. Then there exists a subsequence $\{\mathbf{x}^k\}_{\tilde{\mathcal{K}}}$ such that $\{l(\mathbf{x}^k)\}_{\tilde{\mathcal{K}}} \to \infty$, because $l(\mathbf{x})$ is coercive based on the assumption (A1). Recall that $r(\mathbf{x}) \geq 0$ according to the assumption (A2). Thus, we have $\{f(\mathbf{x}^k)\}_{\tilde{\mathcal{K}}} \to \infty$, which leads to a contradiction with that $f(\mathbf{x}^k) \leq f(\mathbf{x}^0), \forall k \geq 1$. Therefore, $\{\mathbf{x}^k\}$ is bounded, which immediately imply that $\{\mathbf{s}^k\}$ is also bounded. Recalling that $\nabla l(\mathbf{x})$ is Lipschitz continuous with constant, we obtain that $\|\mathbf{y}^k\| \leq L \|\mathbf{x}^k - \mathbf{x}^{k+1}\|$ and hence $\{\mathbf{y}^k\}$ is bounded. Since $-\mathbf{v}^k \in \partial^o f(\mathbf{x}^k)$ and $\{\mathbf{x}^k\}$ is bounded, then based on Proposition 4, we obtain that $\{\mathbf{v}^k\}$ is bounded. $\qquad\square$

Based on Proposition 7, we present the following important properties of L-BFGS.

**Proposition 8** *In the course of the inversion Hessian matrix update using L-BFGS, let $\{H^0\}$ and $\{H^{k-m}\}$ be bounded and positive definite, and $\{\mathbf{x}^k\}$, $\{\mathbf{s}^k\}$, $\{\mathbf{v}^k\}$, $\{\mathbf{y}^k\}$ and $\{\rho^k\}$ be bounded, where $\mathbf{s}^k = \mathbf{x}^{k+1} - \mathbf{x}^k$, $\mathbf{y}^k = \nabla l(\mathbf{x}^{k+1}) - \nabla l(\mathbf{x}^k)$ and $\rho^k = ((\mathbf{y}^k)^T \mathbf{s}^k)^{-1}$. Then there exists a positive constant $M$ such that for all $\mathbf{x} \in \mathbb{R}^n$ and all $k \geq 1$: $\mathbf{x}^T H^k \mathbf{x} \leq M \|\mathbf{x}\|^2$. That is, the eigenvalues of $H^k$ are uniformly bounded from above by $M$. Moreover, $\{\mathbf{d}^k\}$ and $\{\mathbf{p}^k\}$ are bounded.*

**Proof** When $k \leq m$ ($m$ is the unrolling steps of L-BFGS), L-BFGS is equivalent to BFGS and $H^k$ is updated by the recursive relationship in Eq. (29). When $k > m$, $H^k$ is updated by the recursive relationship in Eq. (30). Thus, Eqs. (29), (30) and the boundedness of $\{H^0\}$, $\{H^{k-m}\}$, $\{\mathbf{s}^k\}$, $\{\mathbf{y}^k\}$, $\{\mathbf{v}^k\}$ and $\{\rho^k\}$ immediately imply that $\{\|H^k\|_F\}$ is bounded. That is, there exist an $M > 0$ such that $\|H^k\|_F \leq M$ for all $k \geq 1$. Thus, for all $k \geq 1$, $\lambda_{\max}(H^k) \leq \|H^k\|_F \leq M$, where $\lambda_{\max}(H^k)$ is the largest eigenvalue of $H^k$. That is, there exists a positive constant $M$ such that for all $\mathbf{x} \in \mathbb{R}^n$ and all $k \geq 1$: $\mathbf{x}^T H^k \mathbf{x} \leq M\|\mathbf{x}\|^2$. Thus, the eigenvalues of $H^k$ are uniformly bounded from above by $M$. It easily follows that $\{\mathbf{d}^k\}$ and $\{\mathbf{p}^k\}$ are bounded by noticing that $\{\mathbf{v}^k\}$ is bounded. $\quad\square$

**Remark 6** *We discuss how to guarantee that the conditions in Proposition 8 are satisfied in practical L-BFGS updates. We usually choose $H^0$ and $H^{k-m}$ as multiple identity matrices such that $\{H^0\}$ and $\{H^{k-m}\}$ are bounded and positive definite. Proposition 7 guarantees that $\{\mathbf{x}^k\}$, $\{\mathbf{s}^k\}$, $\{\mathbf{v}^k\}$ and $\{\mathbf{y}^k\}$ are bounded. To guarantee that $\{\rho^k\}$ is also bounded, we adopt a similar strategy presented in [5, 1]: choose a small positive constant $\delta$ and perform L-BFGS updates only when $(\mathbf{s}^k)^T \mathbf{y}^k \geq \delta$.*

**Remark 7** *To guarantee the eigenvalues of $H^k$ are uniformly bounded from below by a positive constant, we can add a small positive diagonal matrix $\nu I$ to $H^k$ (e.g., $\nu = 10^{-12}$). Thus, the eigenvalues of $H^k$ are both uniformly bounded from below by $\nu$ and uniformly bounded from above by $M$, respectively.*