[Reviews · NeurIPS 2015]

Submitted by Assigned_Reviewer_1

This paper presents an algorithm HONOR applicable for a wide range of non-convex sparse learning formulations.

The HONOR method combines a Quasi Newton step with a standard GD step for a first order approximation of the

objective without evaluating the actual Hessian for the QN step and using L- BFGS to scale it up for large scale objectives. The authors show that any limit point the algorithm might converge to is a Clarke critical point of the objective and the sequence generated by the objective leads to a limit point.

The convergence for non-convex problems is typically hard to analyze rigorously.

This paper succeeds in showing the analysis of convergence to a proved limit point which is guaranteed to be a Clarke critical point. This analysis is pretty deep and the mathematics looks correct. It would be good to see whether the same kind of method can be applied to more generalized non-convex objectives.

The empirical evaluation looks pretty reasonable with experiments analyzing the decrease in objective value over time for different large scale and high dimensional data sets. The results show that the algorithm converges faster than other comparable methods. It would be good to see some example of a dataset where there might be a local minima (even if synthetic) and see how the algorithm performs in presence of such an objective as well some guidance to how the different parameters particuarly \gamma can be chosen to ensure faster convergence.
Summary: This paper looks at a specific kind of regularized non-convex problem which appears in different problems in machine learning. They give a hybrid algorithm combining second order information using Quasi newton steps as well as gradient descent steps depending on a certain condition which is cheap to compute at every iteration. They provide an extensive analysis, pointing out the steps that are complicated due to lack of convexity and prove that their algorithm converges to a Clarke critical point. The analysis is also extendable to other non-convex general scenarios.

Submitted by Assigned_Reviewer_2

This paper considers efficient implementations of non-convex sparse learning formulations. Recent studies have shown that many convex sparse learning formulations are inferior to their non-convex counterparts in both theory and practice. However, it is still quite challenging to efficiently solve non-convex sparse optimization problems for large-scale data. This paper presents a novel algorithm HONOR which is applicable for a wide range of non-convex sparse learning formulations.

One of the key ideas in HONOR is to incorporate the second-order information to greatly speed up the convergence, while unlike most existing second-order methods it avoids solving a regularized quadratic programming and only involves matrix-vector multiplications without explicitly forming the inverse Hessian matrix. Thus, HONOR has a low computational complexity at each iteration and it is scalable to large-size problems.

The convergence for non-convex problems is typically challenging to analyze and establish. One major contribution of this paper is to establish a rigorous convergence analysis for HONOR, which shows that convergence is guaranteed even for non-convex problems. The key to the convergence analysis is the hybrid optimization scheme which chooses either a Quasi-Newton step or a Gradient Descent step per iteration. The presented analysis is nontrivial. It will be good to include a high-level description of the intuition behind the proposed hybrid scheme.

The empirical evaluation presented in this paper is convincing. The authors evaluate the proposed algorithm using large-scale datasets which include up to millions of samples and features. Two of the datasets include over 20 million features. Such scale of datasets is needed to evaluate the behavior of the algorithms. Results demonstrate that HONOR converges significantly faster than state-of-the-art algorithms. Some guidance on the selection of \epsilon will be helpful.

It is mentioned that HONOR may have the potential of escaping from high error plateau which often exists in high dimensional non-convex problems. It will be interesting to explore theoretical properties of the HONOR solutions.

The proposed algorithm empirically converges very fast. Is there any guarantee on the (local) convergence rate of HONOR?

Can the proposed algorithm be extended to other sparsity-inducing penalties such as group Lasso and fused Lasso?
Summary: This paper considers efficient implementations of non-convex sparse learning formulations. Recent studies have shown that many convex sparse learning formulations are inferior to their non-convex counterparts in both theory and practice. However, it is still quite challenging to efficiently solve non-convex sparse optimization problems for large-scale data. This paper presents a novel algorithm HONOR which is applicable for a wide range of non-convex sparse learning formulations.

Submitted by Assigned_Reviewer_3

The paper proposes an efficient optimization algorithm (HONOR) for nonconvex regularized problems by incorporating the second-order information to speed up the convergence. Although the convergence analysis for nonconvex optimization methods is typically more difficult than convex methods, the authors have proved that every limit point of the sequence obtained by HONOR is a Clarke critical point. The proposed method is not only theoretically sound, but also computationally efficient. However, I still have a few concerns about the proposed algorithm.

1: Using the fact that each decomposable component function of the non-convex regularizer is only non-differentiable at the origin, the authors have designed an algorithm which can keep the current iterate in the same orthant of the previous iterate so that the segment between any two consecutive iterates do not cross any axis. The strategy seems to make the algorithm dependent on an initial point x^0.

+ Can the authors show how the solution of HONOR is influenced by the choice of the initial solution?

+ Which vector did the authors use as the initial solution in the numerical experiments?

+ Does the results shown in Figure 1 change by using different the initial solution?

2: One of the advantages of HONOR is to incorporate the second-order information to speed up the convergence but HONOR might sacrifice memory usage. When the given problem is highly nonconvex, the positive definite matrix H^k given by L-BFGS might be completely different from the Hessian matrix of the nonconvex optimization problem. Do the authors have any thoughts about those issues?
Summary: The paper is very well-written and an efficient algorithm with theoretical guarantee for nonconvex problems is of great significance. However, I have some concerns about the proposed algorithm.

Submitted by Assigned_Reviewer_4

The paper considers an algorithm for the optimization problem

\min l(x) + r(x)

where l is a smooth function and r(x) = \sum(\rho(|x_i|) for a concave smooth function \rho. The authors present a hybrid algorithm that combines a quasi newton step and a gradient descent step. The main idea of the algorithm is borrowed from the OWL-QN algorithm. That is, in order to deal with the non-smoothness of the regularizer, the iterates of the algorithm are constrained to remain on the same quadrant. However, the fact that the regularizer is non-convex does not allow the use of subgradient properties. Instead, the proofs are based on the properties

Clarke sub-differential.

I believe that the problem being tackled is not trivial. Dealing with non-convexity is an added difficulty to the non-smoothness of the regularizer. There are three problems that I see with this paper.

(1) In spite of the criticism of DC programming for non-convex optimization, this algorithm provides linear convergence guarantees whereas the proposed algorithm does not provide any rates of convergence.

(2) In order to obtain convergence the algorithm requires a gradient descent step. However, it seems that the only way to find the optimal tradeoff between the GD step and the QN step is by actually running the algorithm with different tradeoff parameters.

(3) As a practitioner I have never used a non-convex regularizer like the ones described on this paper nor have I seen them in any publications. Probably this is only because of the type of applications I do but I do want to make sure that this problem is indeed relevant to the machine learning community.

Summary: The paper deals with minimization

under a non-convex regularizer that promotes sparsity. The paper is well written and the math is correct. However I am not entirely sure of the relevance of the results for the learning community.

Author Feedback
Author rebuttal: Thanks for all reviewers' comments.

To Reviewer 1

Regarding the intuition behind the proposed hybrid scheme, we have presented a high-level description on Lines 168-172 (page 4).

Regarding the guidance on the selection of \epsilon, we usually set \epsilon as a very small value, e.g., \epsilon=1e-10 usually works very well. Experimental results also show that a very small \epsilon is always safe to guarantee the fast convergence of HONOR.

The suggestions on the theoretical properties of the HONOR solutions, local convergence rate analysis and potential extensions to other sparsity-inducing penalties are very interesting research directions in the future.

To Reviewer 2

Regarding the dependence on an initial point x^0, we want to clarify that it is the non-convexity of the problem that makes the algorithm dependent on an initial point (If the problem is convex, the algorithm will be independent of the initial point). It is a common issue that a non-convex optimization solver may be dependent on the initial point. Our results demonstrate that HONOR can produce a solution with reasonable quality in the sense that the final objective function value is usually smaller compared with other algorithms (e.g., GIST).

Regarding the setting of initial point in the numerical experiments, we use a random vector whose entries are i.i.d. sampled from the standard Gaussian distribution, which was stated on Line 368.

Regarding the results by using different initial solutions, we run the algorithms many times with different random vectors as initial points and find that HONOR converges very fast, although the final solutions may be different.

Regarding the positive definite matrix H^k given by L-BFGS in highly non-convex problems, we want to clarify that this is a common issue in non-convex problems. For a non-convex problem, the Hessian matrix is not positive definite. Thus we cannot use it directly even if we can obtain the exact Hessian matrix easily. One possibility is to use an approximate positive definite matrix instead of the Hessian matrix to capture the second-order information.

To Reviewer 3

Regarding the rates of convergence for the proposed algorithm, we want to clarify that it is generally difficult to conduct convergence analysis for non-convex problems. Providing a convergence rate analysis for non-convex problems is much more challenging. But this is an interesting future work we can explore.

Regarding the optimal tradeoff between the GD and QN steps, we practically set \epsilon as a very small value (e.g., 1e-10), which usually works very well. Experimental results also show that a very small \epsilon is always safe to guarantee the fast convergence of HONOR. So in practice, no parameter tuning is needed.

Regarding the relevance of the non-convex penalties to the machine learning community, we want to clarify that the non-convex penalties have been widely used in machine learning, since they were proposed in [6, 10, 23]. By searching the above three papers in google scholar, we can find that they have been cited by many machine learning related papers. Due to the word limit of the feedback, I only list a few here: (1) J Fan, R Samworth, Y Wu. Ultra-high dimensional feature selection: beyond the linear model. Journal of Machine Learning Research, 2009. (2) F Bach, R Jenatton, J Mairal, G Obozinski. Optimization with Sparsity-Inducing Penalties. Foundations and Trends in Machine Learning, 2012. (3) P Loh, M Wainwright. Regularized M-estimators with nonconvexity: Statistical and algorithmic theory for local optima. NIPS, 2013. (4) M Kolar, J Sharpnack. Variance function estimation in high-dimensions. ICML, 2012.

To Reviewer 4

Thanks for the comments.

To Reviewer 5

We want to clarify that the proposed HONOR algorithm and its convergence analysis are non-trivial in several aspects: (1) HONOR is not a trivial combination of quasi-newton and gradient descent steps. The gradient descent is only introduced when some "extreme cases" happen [see Lines 168-172, page 4]. The combination strategy is motivated by the convergence guarantee of the algorithm. (2) The major body of the HONOR algorithm and convergence analysis focuses on the quasi-newton step, which is highly non-trivial and does not rely on the gradient descent step. (3) The optimization problem is non-convex, which makes the traditional convergence analysis techniques (e.g., subdifferential) in convex problems not applicable here. So we use the Clarke subdifferential to characterize the optimality of problem. Due to the lack of convexity, some basic properties of the subdifferential of a convex function may not hold for the Clarke Subdifferential of a non-convex function, e.g., the subgradient and the directional derivative may not exist for non-convex problems.